# Optimization of Air Backwash Frequency during the Ultrafiltration of Seawater

**DOI:** 10.3390/membranes10040078

**Published:** 2020-04-22

**Authors:** Clemence Cordier, Tarik Eljaddi, Nadjim Ibouroihim, Christophe Stavrakakis, Patrick Sauvade, Franz Coelho, Philippe Moulin

**Affiliations:** 1Aix Marseille Univ, CNRS, Centrale Marseille, M2P2-EPM (UMR 7340), 13545 Aix en Provence, France; clemence.cordier@univ-amu.fr (C.C.); eljaddi@gmail.com (T.E.); nadjim.ibouroihim@etu.univ-amu.fr (N.I.); 2Plateforme Expérimentale Mollusques Marins, Station Ifremer de Bouin, Polder des Champs, 85230 Bouin, France; christophe.stavrakakis@ifremer.fr; 3Suez—Aquasource, 20, Avenue Didier Daurat, CEDEX 04, 31029 Toulouse, France; patrick.sauvade@suez.com (P.S.); franz.coelho@suez.com (F.C.)

**Keywords:** ultrafiltration, seawater treatment, air backwash, fouling control

## Abstract

The main objective of this paper is to study the effect of new air backwash on dead-end ultrafiltration of seawater with a pilot at semi-industrial scale (20 m^3^/day). To control membrane fouling, two different backwashes were used to clean the membrane: classical backwash (CB) and new air backwash (AB) that consists of injecting air into the membrane module before a classical backwash. To evaluate the efficiency of AB and CB, a resistance in series model was used to calculate each resistance: membrane (R_m_), reversible (R_rev_) and irreversible (R_irr_). The variation of the seawater quality was considered by integrating the turbidity variation versus time. The results indicate clearly that AB was more performant than CB and frequency of AB/CB cycles was important to control membrane fouling. In this study, frequencies of 1/5 and 1/3 appear more efficient than 1/7 and 1/9. In addition, the operation conditions (flux and time of filtration) had an important role in maintaining membrane performance—whatever the variation of the seawater quality.


**Highlights:**
New air backwash in two stepsNew method based on turbidity integration to check the air backwash/backwash efficiency.


## 1. Introduction

Nowadays, clarification and disinfection of water is very important before using or reusing of water in many applications: aquaculture, agriculture irrigation, industrial use, urban green surface [1,2,3]. It is necessary to ensure a constant quality water, either at input or output of processes. Conventional water treatment processes are based on several stages, that take more time, have a high footprint on environment, and do not ensure a constant quality of treated water under the variable operating conditions [4,5]. However, ultrafiltration (UF) can clarify and disinfect water [6] more especially for drinking water production from natural water. It can be used for seawater or underground water [7,8] as a barrier to viruses and bacteria. Ultrafiltration operates at low pressure (0.2 to 2 bars) and the operational and capital expenditures can be low in comparison with other process [9]. The fouling of membrane can affect the performance of the process [10] and for this reason, many strategies are developed to reduce/remove fouling by using a backwash, air backwash or chemical cleaning which is function of the quality of feed water [11,12]. Backwash consists to recycle clean water from permeate side to feed side, to remove reversible part of the fouling by physical effect of water or a mixture of water/air. The chemical cleaning is used when the irreversible fouling becomes so important. Several studies show that air/water backwash is called by many different names, such as air sparging, air/water flushing, air lifting, air scrabbling, air scouring and gas backwash [12,13]. It is proved that this type of backwash including air/water is more efficient than normal backwash because the air help to dislodge the cake of fouling [4,14]. In our knowledge, all papers discuss the use of air while filtration cycle (injection in feed) or during backwash by injection of air simultaneously in water [12]. In this case, three kinds of water and air flows can be distinguished: a bubble flow where the air is dispersed in water phase, a slug flow with a succession of gas slugs followed by liquid slugs, and an annular flow where a continuous air flows in the center of pipe or fiber [14]. In this paper, a semi-industrial UF pilot was used to study the efficiency of new air backwash in two steps (i.e., the novelty of this paper) and its frequency during the seawater treatment: Air backwash (AB) consists (i) to inject air in membrane module before (ii) a classical backwash. In the case of specific effluent this new backwash shows high performances [15,16]. In this paper, the ultrafiltration of seawater, generally used as pretreatment, is operated. The efficiency of the new air backwash is studied and the optimization of the frequency of this air backwash (AB) is realized using resistance in series model by integrating the quality variation of the seawater versus time.

## 2. Permeability and Resistance

Ultrafiltration is a pressure driven process where the flux *J* (L·h^−1^·m^−2^) is calculated by the product between the membrane permeability and the transmembrane pressure, TMP (bar):(1)J=Lp TMP

It is important to note that the ultrafiltration is conducted at constant flux (*J*). In the case of pure water, the permeability can be calculated as the function of the membrane resistance (*R**_m_*) and the viscosity of the water at operating temperature:(2)$$L_{p0}=\frac{1}{\mu R_{m}}$$
where *R_m_*: membrane resistance (m^−1^), µ: seawater viscosity (Pa s). Lp_0_ is initial permeability of membrane (L h^−1^ m^−2^ bar^−1^). For the membrane used, the initial permeability is equal to 880 L h^−1^ m^−2^ bar^−1^ and the membrane resistance is 4.1 × 10^11^ m^−1^. During the filtration of seawater, after backwash, the initial permeability of membrane is not totally restored because an irreversible fouling is still on membrane, corresponding to an irreversible resistance (*R_irr_* (m^−1^)) that can be calculated by (3):(3)$$Lp=\frac{1}{µ\left(R_{m}+R_{irr}\right)}$$

At the end of a filtration cycle and before the backwash, another resistance must be added, *R_rev_* (m^−1^), corresponding to reversible fouling that can be removed by backwash. At this moment, the permeability is function of three resistances (*R_m_*, *R_irr_*, *R_rev_*), that can be expressed by the equation (4):(4)$$Lp=\frac{1}{µ\left(R_{m}+R_{irr}+R_{rev}\right)}$$

Filtration cycle after cycle, the variation of reversible and irreversible resistance can be evaluated. The fouling was not the same for a given volume of permeate due to the quality of the seawater which is variable with time. The concentration of suspended matter is proportional to the turbidity that is measured in real time and the integration of this parameter versus time will take in consideration the quality of the seawater for each filtration cycle. The addition of all these integrals will give the total variation of turbidity between 2 chemical washes. It is expressed by the relation:(5)$$\sum \int_{i+\left(n-1\right)}^{i+\left(n+1\right)}Turbidity\;dt\;(NTU.min)$$ where *i*: filtration cycle.

## 3. Material and Methods

All experiments were conducted using a hollow fiber polyether sulfone membrane provided from SUEZ-AQUASOURCE with filtration from inside of fiber to outside with initial permeability of 880 L h^−1^ m^−2^ bar^−1^, module area 8 m² and a pore size of 20 nm. The pilot plant used in this study is a semi industrial scale and it is completely automated; it can treat 20 m^3^.day^−1^ (Figure 1). The system is programmed to adapt the filtration parameters to the quality of feed water. All experiments were carried out with dead-end filtration mode, at a constant flux rate (J) and the feed seawater was pretreated by a prefilter with 130 µm. Over one year of experiment, the minimum-average-maximum values of pH and salinity of the water supplying the membrane are, respectively 7.3–7.9–9.4 and 31.4–33.9–35.3 mg L^−1^. The turbidity parameter is more affected by seasonal variation with a range between 1 and 50 NTU. The pilot plant can clean membrane module automatically and the parameters can be changed by the operator. The system includes the classical backwash (CB), air backwash (AB) and chemical cleaning (CEB). As mentioned here before, the CB is the injection of permeate from permeate side to feed side (inverse of filtration mode) with a flow rate of 2.5 m^3^ h^−1^ [15,16] and AB air injection inside fibers to discharge water to outside until to drain all fibers when the pressure reached 0.3 bar. The air is supplied with an air compressor. This operation is followed immediately by a classical backwash. Finally, when the CB and AB were not efficient to restore the initial permeability before to start with new operating conditions, the CEB was conducted with a mixture of (soda + chlorine) and acid.

The feed seawater used is located at Ifremer Station, Bouin (Bouin, France) for which seawater received several pretreatments (sand filtration (25–30 µm) and UV). Turbidity of feed was recorded online every minute by VisioTurb 700IQ SW WTW (Weiheim, Germany). The objective was to optimize the hydraulic parameters like frequency of AB to control the membrane fouling. Two cases were studied, the first one was the variation of the frequency of AB (AB/CB frequency 1/3 1/5 1/7 1/9) for a constant couple of permeate flux and filtration time (J = 60 L m^−^² h^−1^ and t_filtration_ = 60 min ) and the second one, for a constant frequency (AB/CB = 1/5) variation of the couple of permeate flux and time filtration was studied. Data, permeate flux, temperature, TMP, turbidity were recorded automatically by UF pilot every 1 min. The minimum and maximum durations tested during this study were, respectively 2600 and 29,900 min.

## 4. Results and Discussion

### 4.1. Air Backwash and Classical Backwash

Filtration performance was recorded every minute to study the impact of classical backwash (CB) and air backwash (AB) on hydraulic performances. For each couple (*J*; t_filtration_), the variation of permeability and turbidity can be drawn. As expected, even with the pretreatment, fouling is still severe for the dead-end UF membrane used during more than 9000 min. A significant decrease of the permeability is observed when the turbidity increases over the duration of filtration, as shown in Figure 2 with J = 60 L h^−1^ m^−2^ and t_filtration_ = 60 min. It is important to note that the fouling is severe, as for large scale desalination plants, with a backwash each hour and 2 chemical washes per day to try to limit this decrease: the variation of permeability versus time explains this severe fouling. As we can see, the variation of the membrane performance is affected by the turbidity variations. A strong increase in the turbidity of the inlet water impacts the decrease in permeability with a steeper slope (red line on Figure 2). Of course, the fouling of the membrane is not only due to the suspended matter (i.e., organic matter), but this parameter seems to control the hydraulic performances. About the membrane performance, a turbidity lower than 1 NTU is obtained in the permeate. As expected, the pH and salinity is not affected by ultrafiltration: over one year of experiment, the minimum-average-maximum values of pH and salinity of the UF permeate are, respectively 7.3–7.9–9.1 and 31.5–34.0–35.7 mg L^−1^. Moreover, when the filtration time increases, the permeate flow decreases due to the accumulation of material (membrane fouling) versus time. When the permeate flow is fixed, the TMP increases (driving force) and the permeability therefore decreases over time.

To eliminate this fouling, the effectiveness of AB was compared to that of CB. In order to quantify this benefit, the gain in TMP (in mbar) after backwash was measured for the different conditions studied. For a constant frequency (AB/CB= 1/5), the gain in TMP after AB was compared to the average of the gains of the previous 5 CB. The results obtained for the different filtration conditions tested are presented in Figure 3. As mentioned before, a severe fouling appears, so the duration of filtration step is adapted as the function of the permeate flux. When the filtration time increases the duration decreases otherwise, we will have an irreversible fouling even by chemical wash. This explains why the filtration times studied are not the same depending on the permeate flux. It is important to note that all this study is done, at a constant flow rate, on real seawater that varies over time and these results have always been observed during all experiments with different qualities of feed water but a similar trend. It appears that the TMP improvement is greater with AB than with CB. Air backwash will get better performance than the CB, because for a constant permeate flow rate after the AB, the TMP is always lower than after a CB (i.e., the removal of the fouling is more important). If this TMP parameter is important for the ultrafiltration monitoring process and the benefits in mbar thus recovered is essential for operation cost, it is more difficult to quantify. Hence, Chang et al. [4] reported several methods in the literature to quantify the effect of backwash: for a constant flux condition, it is better to use the relationship (5) proposed by Chellam [17] that consider the variation of transmembrane pressure before and after backwash:(6)$$\eta \;\left(\%\right)=100\frac{\left(P_{f}-P_{0}\right)}{\left(Pf-Pi\right))}$$
with, ƞ: cleaning efficiency (%), *P_f_*: pressure at the end of the filtration cycle (bar), *P_0_*: pressure after backwash (bar), *P_i_*: pressure at the start of the filtration cycle (bar).

The value of efficiency, less telling in operation, brings up a notion which must be explained but not discussed by Chellam et al. or Waterman due to the studies carried out [17,18]: an efficiency higher than 100% can be obtained. This result reflects a better backwash efficiency than the previous one. Figure 4 shows an efficiency greater than or equal to 100% for AB, while this is less than or equal to 100% for CB. When fouling increases, generated by an increase in flow rate and / or filtration time, the TMP benefits increases for CB and AB. However, regardless of filtration conditions, air injection improves cleaning efficiency. For the same permeate flux of 60 L h^−1^ m^−2^ and for different filtration times from 20 to 60 min, the regeneration of the membranes was improved between 16% and 42%. Similar results were obtained for a constant filtration time and different permeate flux. These results establish the effectiveness of AB on removing fouling generated by seawater whatever the quality of the feed seawater and the operating conditions.

### 4.2. Effect of Air Backwash Frequency

To check the effect of air backwash frequency, the irreversible resistance (*R_irr_*) is dimensionless. The irreversible resistance was divided by membrane resistance and this ratio was divided also by the initial ratio of (*R_irr_*/*R_m_*) _initial_ at the start of cycle. The irreversible resistance is calculated according to resistance in series model by taking in consideration the notion of irreversibility for backwash and air backwash. It was important to check if the representation (*R_irr_*/*R_m_*)/(*R_irr_*/*R_m_*)_ini_ = f(∑∫NTU) gives similar results whatever the qualities of water with constant operating parameters. The experiments were repeated several times, for different qualities of seawater, for different frequencies of AB/CB. It should be noted that for these results, the couple *J* = 60 L h^−1^ m^−2^; t_filtration_ = 60 min is used, this means that a point on the curve will represent 60 minutes of data points (more than 420 data). At the end of these experiments, whatever the AB frequency and operating conditions, a good reproducibility was obtained whatever the variation of the feed seawater quality in terms of turbidity (Figure 5).

The fact that whatever the water quality, the variations are similar shows that the calculation of the ∑∫NTU seems realistic to consider the load of the water to be treated. Figure 6 represents the variations of (*R_irr_*/*R_m_*)/(*R_irr_*/*R_m_*)_ini_=f(∑∫NTU) for the different AB / CB frequencies tested and the same couple of flux and filtration time. It should be noted that for each frequency, at least two experiments reflecting two different qualities of water are represented.

The influence of air backwashing on the elimination of fouling is almost similar for 1/3; 1/5 and 1/7; 1/9 frequencies. When the air backwash frequency increases (1/3 and 1/5), the regeneration step is more effective to clean membrane because it can remove more fouling and the irreversible resistance increases slowly. Conversely, when the air backwash frequency is low, the irreversible resistance will be accumulated, and the membrane cleaning will be less effective (1/7 and 1/9). This result is similar to what is mentioned in another paper [19]. The shape of the curves, "exponential" for frequencies 1/7 and 1/9 and "linear" for frequencies 1/3 and 1/5, shows the efficiency of this frequency on the filtration of seawater. The notions of exponential and linear curves are only subjective and unrealistic since the presence of AB in front of CB brings great variations as shown in the left part of Figure 7d. The linear variation reflects the fact that the AB allows to maintain a proportionality between the irreversible resistance and the quantity of matter at the membrane. On the other hand, the exponential form shows that this accumulation of matter adds an intensification of the fouling which can be caused for example by a densification of the cake by the fine particles and / or the organic matter displayed in seawater that the AB does not eliminate effectively due to its less frequency. In the literature, Ye et al. (2011) [20]. observed that the thickness of the cake increased with the consecutive cycles of filtration / backwashing despite constant resistance over the duration of filtration. This observation suggests that no change in the structure of the cake over time is operated (only the thickness). In addition, the authors showed that the resistance values resisted with the number of backwash cycles, reflecting a decrease in the effectiveness of these cleaning procedures. The approach developed can be discussed since the permeability of the membrane at the start of cycles (after a chemical backwash) can present small deviation of the order of 150 L h^−1^ m^−2^ bar^−1^ from one cycle to another even if it was shown in previous work [16] that the permeability of the membrane remained constant after the chemical cleanings during more than 1 year. In order to study the evolution of fouling over time, the results were studied considering the three resistances of transfer (membrane, reversible and irreversible).

In the case of the frequency AB/CB 1/3, Figure 7a, the shape of the irreversible resistance can be divided into two parts: (1) it is a line that describes the proportional elimination of the cake during filtration. The reversible resistance is not constant but rather increases at a slower pace than *R_irr_*. The resistance of the membrane represents from 4/7 to 4/8 of the total resistance. (2) is an exponential which shows an increasingly removal difficulty of fouling with a very strong increase of the irreversible resistance due to an accumulation of compounds on the membrane surface cycle after cycle. This accumulation creates a high density of the cake. It is then noted that the reversible resistance is also impacted, and the membrane resistance becomes minority at the end (4/13 of the overall resistance). These similar two parts were found in the case of AB/CB at 1/5 (Figure 7b) with a first part where the transfer resistance is mainly composed by resistance of membrane. The two resistances, reversible and irreversible, increase slightly. For case AB/CB at 1/7 (Figure 7c), this phenomenon is strongly accentuated with a fast and exponential variation of the two resistances compared to the accumulated matter. When the frequency of AB/CB is 1/9 (Figure 7d), there is a very significant increase at the beginning. In this condition, the benefit of AB is also clearly visible but arrives too late to maintain a low fouling. This study on the frequency of AB made it possible to study the behavior of air backwashes on the ultrafiltration performances taking into account the impact of turbidity (i.e., quality of water). The frequency AB/CB of 1/5 was evaluated as the most suitable because it was the most efficient condition with the frequency 1/3, but less stressful for the fibers and less consuming permeate (10%). In the case of a constant AB/CB frequency with a variation of the operating parameters (*J*; t_filtration_), the main results are presented in Figure 8.

The longer the filtration time, the greater the drop in permeability obtained after each backwash. This is consistent with the fact that the filtration time proportionally reflects the volume of permeate passed through the membrane at constant filtration flow. If we compare the data with *J* = 80 L h^−1^ m^−2^; t_filtration_ = 20 min and *J* = 60 L h^−1^ m^−2^; t_filtration_ = 30–45–80 min, the filtered permeate volumes are, respectively 213–240–360–640 L. Despite a lower volume of permeate, for *J* = 80 L h^−1^ m^−2^, the permeability strongly decreases which can be explained by a very high permeate flux creating an important fouling. This result is in accordance with Waterman [18] study, at the higher operating flux, the fouling rate increased because the rate of approach of foulants to the membrane surface increased.

## 5. Conclusions

The filtration of seawater at semi-industrial scale has highlighted the improvement of membrane cleaning performance with novelty air backwash (AB) in two steps compared to classical backwash (CB). Whatever the quality of the feed seawater and the operating conditions, the recovery of initial permeability is better with air (AB) than without (CB). In comparison with other papers, it is the first time that the effect of backwash is quantified higher than 100%, due to the injection of air before. It is the first time that integration of the turbidity versus time, to take into account the quality of seawater, is used. With this methodology, it is possible to compare AB or CB efficiency because the variation of the seawater quality and its impact on the membrane fouling is considered. For constant operating conditions but for different seawater qualities, the variation of the irreversible fouling (i.e., the variations of (*R_irr_*/*R_m_*)/(*R_irr_*/*R_m_*)_ini_) was the same. The results obtained about the impact of the frequency of AB on the reversible and irreversible fouling of the membranes, show that (i) a frequency of 1/3 or 1/5 limits the membrane fouling and at the opposite a frequency of 1/7 or 1/9 is not enough to maintain a low fouling and in this case a chemical wash was necessary. (ii) AB frequency 1/5 is the most efficient and allows to maintain the performance of the membrane over long periods of time due to a low water consumption. Furthermore, for a same frequency, the operation conditions are very important, and the optima parameter must be selected to maintain the performance of membrane. Finally, it should be noticed that AB can improve the performance of membrane a longer time than CB, consequently the period between two chemical backwash (CEB) will be long and the process will be more economically and environmentally friendly by reducing the use of chemical products. The energy consumption for the AB, not estimated, is very low because only a volume of 20 L (volume of membrane) at 0,3 bar of air is used per each AB.

## Figures and Tables

**Figure 1 membranes-10-00078-f001:**
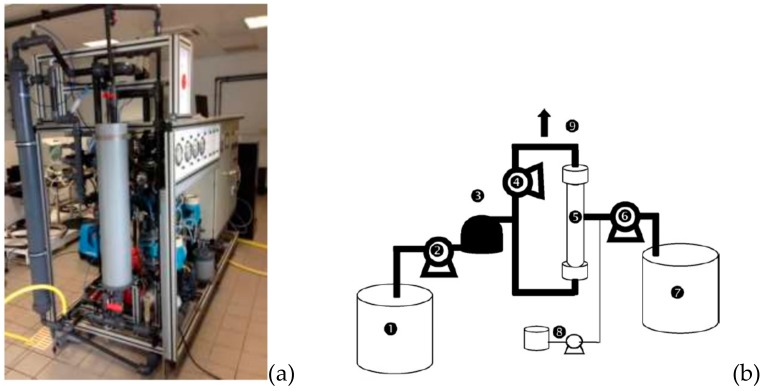
(**a**) Picture and (**b**) scheme of ultrafiltration (UF) pilot. 1: feed tank; 2: feed pump; 3: pre-filter; 4: recirculating pump; 5: membrane module; 6: backwashing pump; 7: purified water tank for backwash; 8: chemical cleaning part; 9: purge}.

**Figure 2 membranes-10-00078-f002:**
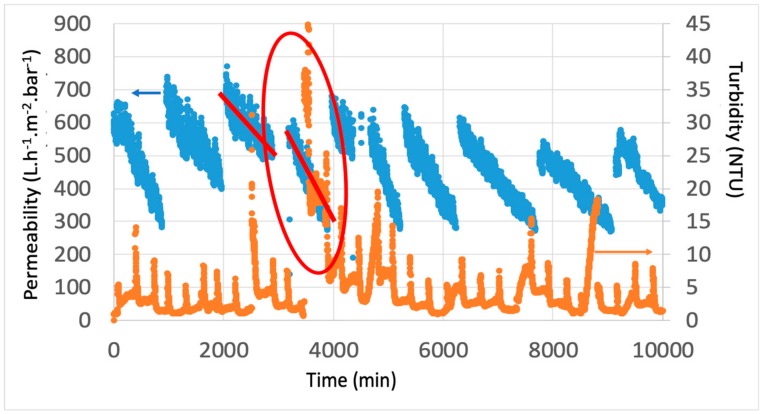
Variation of membrane permeability and turbidity versus time (air backwash (AB)/ classical backwash (CB) frequency: 1/3; *J* = 60 L h^−1^ m^−2^ t_filtration_ = 60 min).

**Figure 3 membranes-10-00078-f003:**
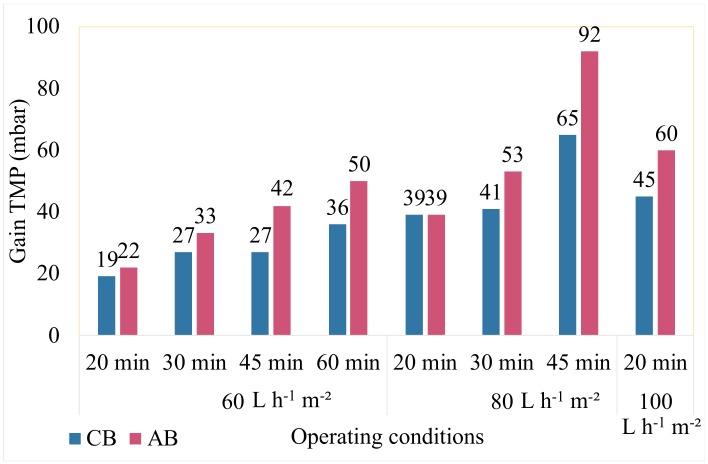
Benefits in PTM after CB and AB versus the operating conditions. Seawater filtration (AB/CB frequency = 1/5).

**Figure 4 membranes-10-00078-f004:**
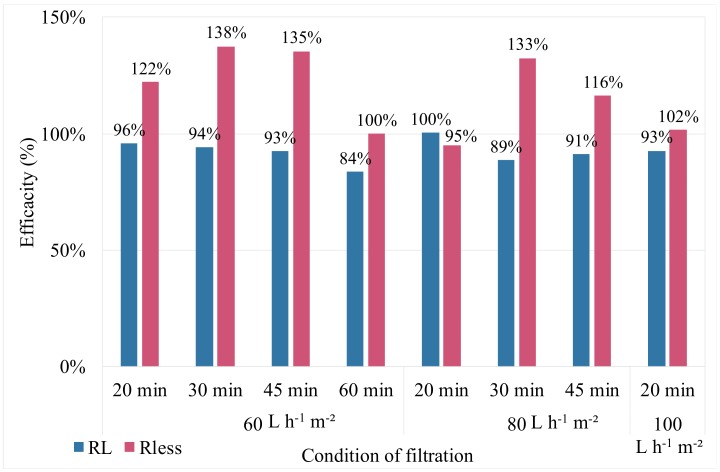
Efficiency of CB and AB versus the operating conditions. Seawater filtration (AB/CB frequency = 1/5).

**Figure 5 membranes-10-00078-f005:**
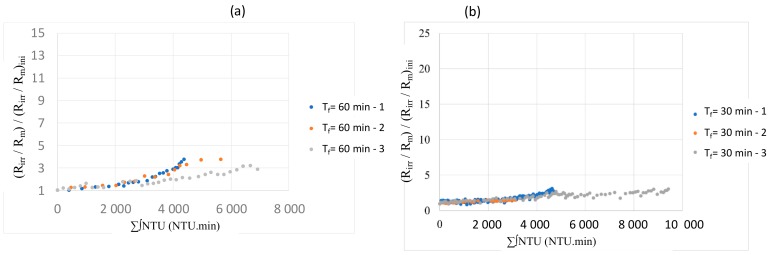
Variation of irreversible resistance versus the total of the integral of turbidity for different filtration durations (AB/CB frequency 1/3; J = 60 L h^−1^ m^−2^). (**a**) t_filtration_ = 60 min and (**b**) t_filtration_ = 30 min.

**Figure 6 membranes-10-00078-f006:**
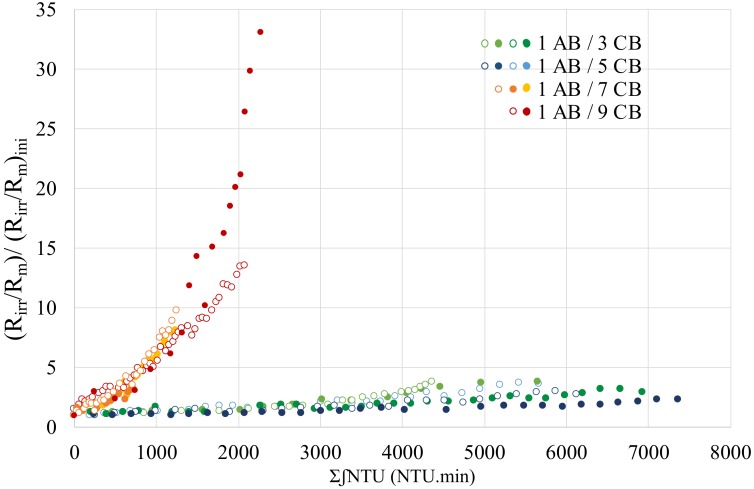
Variation of irreversible resistance versus the total of the integral of turbidity for different AB/AC frequencies (*J* = 60 L h^−1^ m^−2^ and t_filtration_ = 60 min).

**Figure 7 membranes-10-00078-f007:**
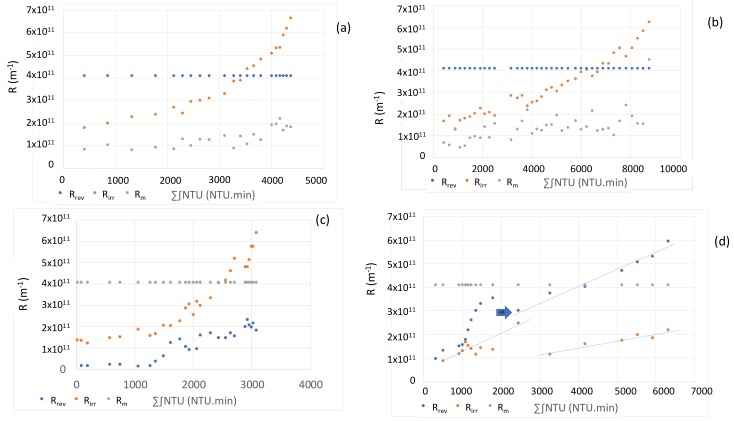
Variation of resistances (irreversible, reversible, membrane) versus the total of the integral of turbidity for different frequencies of AB/CB: (**a**) 1/3, (**b**) 1/5, (**c**) 1/7 and (**d**) 1/9.

**Figure 8 membranes-10-00078-f008:**
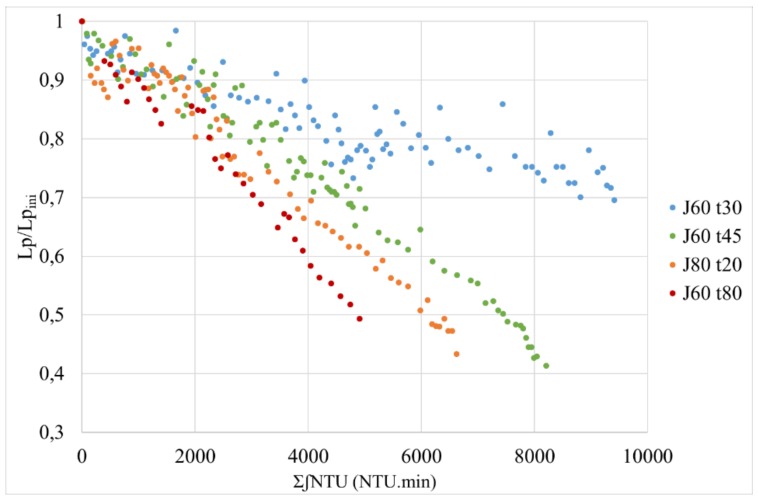
Evolution of permeability versus the total of the integral of turbidity for frequency AB/CB= 1/5 for different conditions of filtration (flux and filtration duration).

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
