# Peer review of "Optimization of Air Backwash Frequency during the Ultrafiltration of Seawater"

_membranes, 2020, doi:10.3390/membranes10040078_

Round 1

Reviewer 1 Report

In this work, the author reported the use of air backwash to minimise the fouling on a dead end ultrafiltration pilot unit. The concept is interesting. However, the manuscript is poorly presented and the data was poorly explained . Therefore, I do not recommend the publication of this manuscript in Membrane Specific comments are listed as follows: 1. In the introduction, the authors claimed that “UF can clarify and disinfect water in one step” this is not true. Many organic compounds/viruses that has molecular weight size smaller than UF are able to pass thru the UF filter, e.g HA. 2. Line 39: AB has not been defined. 3. The explanation on the experimental setup/protocol is not very clear. The item in Figure 1 should be labelled using English. 4. Can the aurther how the air is supplied? Through air compressor? What is the flow rate of the backwash air? 5. Please check reference throughout the manuscript. “Error! Reference source not found” appears many times throughout the manuscript eg. Line 82, 140 6. Check the Figure number throughout. The figure “variation of membrane permeability and turbidity versus time” should be Figure 2. 7. Can the author estimate the energy consumption for the air backwashing? 8. It is unclear the meaning of gain TMP? Is it the amount of TMP reduced after backwashing? 9. The Figure in the manuscript is totally messed up. Pg 10 there appears to be another Figure 2d. This make the manuscript very difficult to understand and follow. Furthermore, the English structure needed significant improvement.

Author Response

Referee 1

Thank you very much for this report

In this work, the author reported the use of air backwash to minimise the fouling on a dead end ultrafiltration pilot unit. The concept is interesting. However, the manuscript is poorly presented and the data was poorly explained.

Thank you, we hope that all these corrections, in light of the remarks of the referees, can make the manuscript publishable in membrane

In the introduction, the authors claimed that “UF can clarify and disinfect water in one step” this is not true. Many organic compounds/viruses that has molecular weight size smaller than UF are able to pass thru the UF filter, e.g HA.

We have written this sentence from a reference [6]. In agreement with your comment we have modified this sentence

Line 39: AB has not been defined.

We are agreed with you, it is not clear. We have removed this notation “AB” (line39) and defined AB line 47

The explanation on the experimental setup/protocol is not very clear.

We have simplified this experimental setup protocol because this protocol is already described in other references [14, 21]. We have modified Figure 1 to improve the filtration setup

The item in Figure 1 should be labelled using English.

We have translated this figure

Can the author how the air is supplied? Through air compressor? What is the flow rate of the backwash air?

Yes the air is supplied by an air compressor. A sentence was added. As for an industrial plant the air pressure is regulated and not the flow rate to prevent a membrane damage. The value of the pressure is given in the text.

Please check reference throughout the manuscript. “Error! Reference source not found” appears many times throughout the manuscript eg. Line 82, 140

We have checked the references and for us there is no error

Check the Figure number throughout. The figure “variation of membrane permeability and turbidity versus time” should be Figure 2.

We have carefully checked the figure number in agreement with your comment

Can the author estimate the energy consumption for the air backwashing?

No, the energy consumption for the air backwashing is not estimated but it is very low. Only an air volume (20L) (volume of fiber) is used and a similar quantity of water. We have added a sentence.

It is unclear the meaning of gain TMP? Is it the amount of TMP reduced after backwashing?

We worked at a constant flow rate so when the fouling decrease, the TMP was reduced. The notion of constant flow rate was missed in the paper and in agreement with your remark, we have added this information and we have added an explanation in the result part.

The Figure in the manuscript is totally messed up. Pg 10 there appears to be another Figure 2d. This make the manuscript very difficult to understand and follow. Furthermore, the English structure needed significant improvement.

A native speaker read and corrected this paper

Reviewer 2 Report

I have reviewed carefully the manuscript. Frankly speaking, the manuscript was poorly organized and written. I strongly advise its publication in Membranes.

First of all, the use of dead-end UF membrane system for seawater treatment is NOT practical at all. Fouling is very severe and flux declined rapidly with the use of dead-end membrane. I don't know how the authors could produce convincing data (Figure 1) as a function of time since feed solution concentration is getting higher and higher as time proceeds. Besides, there is no indication for the data in Orange and Blue (Figure 1).

All the figures are in fact not professionally prepared. Most of them are low resolution and unclear. Besides, there are 4 images in Figure 5, but there only 2 images (a and b) are described in the caption.

There are several figures between Figure 6 and 8, but they come without figure number and caption!!

The selection of 200-kDa PES membrane for pretreatment is unclear. In most of the large-scale seawater desalination process, membrane with much smaller pore size is used!

Throughout the manuscript, there is no any instrumental characterization on the membrane fouling. Such characterization is compulsory for high-quality paper publication.

The properties of feed solution to PES membrane should be provided. Authors also need to provide what are the pretreatment processes conducted on the seawater before it was used for hollow fiber membrane treatment.

The separation data of PES membrane is completely missing! Such data is critical to correlate to the surface fouling!

Author Response

Referee 2 : I have reviewed carefully the manuscript. Frankly speaking, the manuscript was poorly organized and written. I strongly advise its publication in Membranes.

Thank you, we hope that all these corrections, in light of the remarks of the referees, can make the manuscript publishable in membrane

First of all, the use of dead-end UF membrane system for seawater treatment is NOT practical at all. Fouling is very severe and flux declined rapidly with the use of dead-end membrane. I don't know how the authors could produce convincing data (Figure 1) as a function of time since feed solution concentration is getting higher and higher as time proceeds. Besides, there is no indication for the data in Orange and Blue (Figure 1).

We are agreed with your comment. In fact we have specified in our paper that The seawater effluent used is located at Ifremer Station, Bouin (France) for which seawater  received several pretreatments. We have added the detail of these pretreatments used (sand filtration and UV). We have modified the figure 1 to specify the orange and blue curves

All the figures are in fact not professionally prepared. Most of them are low resolution and unclear. Besides, there are 4 images in Figure 5, but there only 2 images (a and b) are described in the caption.

There is a mistake, only 2 images on figure 5???

There are several figures between Figure 6 and 8, but they come without figure number and caption!!

We have modified all the figures in agreement with your remarks

The selection of 200-kDa PES membrane for pretreatment is unclear. In most of the large-scale seawater desalination process, membrane with much smaller pore size is used!

Yes and no. Yes-We are agreed with your remark and now the pretreatment was defined and No- It is function of the pretreatment and it is for desalination (no our case). We have already studied the comparison between MF and UF for seawater filtration for other application and 20 nm of pore size is perfect.

Guilbaud J., Wyart Y., Moulin P. “ Economic viability of treating ballast water of ships by ultrafiltration as a function of the process position” Journal of Marine Science and Technology 12 (2018) 1-12 (DOI: 10.1007/s00773-018-0618-3)

Guilbaud J., Y. Wyart, K. Klaag and P. Moulin, Comparison of seawater and freshwater ultrafiltration on semi-industrial scale: ballast water treatment application, Journal of membrane science and research 4 (2018) 136-145

This pore size is now given in the membrane characteristics

Throughout the manuscript, there is no any instrumental characterization on the membrane fouling. Such characterization is compulsory for high-quality paper publication.

It is impossible to characterize the membrane fouling because we used seawater with a high variation of the quality of this seawater. Moreover, the flow rate is very important. It is not a study at a lab scale.

The properties of feed solution to PES membrane should be provided. Authors also need to provide what are the pretreatment processes conducted on the seawater before it was used for hollow fiber membrane treatment.

In agreement with your comment we have added these informations

The separation data of PES membrane is completely missing! Such data is critical to correlate to the surface fouling!

We have added this information.

Reviewer 3 Report

This manuscript studied the effect of air backwash on the UF performance. It is important for the usage of UF process. However, some issues within the manuscript need to be addressed.

  1. The introduction should be improved. State of the art of the studies about UF backwash should be described more clearly. Please present the novelty of this paper as compared with those published papers about the relevant studies.
  2. Page 2. This section is not “Theory”. It can be a part of experimental and description of Lp should be added. Eq. (5) is not an Equation.
  3. Please explain why air backwash will get better performance than the CB?
  4. The figures should be presented and numbered in sequence.

Author Response

Referee 3: The introduction should be improved. State of the art of the studies about UF backwash should be described more clearly. Please present the novelty of this paper as compared with those published papers about the relevant studies.

We have modified the introduction to put in light the novelty of our paper

Page 2. This section is not “Theory”. It can be a part of experimental and description of Lp should be added. Eq. (5) is not an Equation.

We have modified the section title and the other point

Please explain why air backwash will get better performance than the CB?

We have added an explanation in agreement with your comment

The figures should be presented and numbered in sequence.

The figure was presented and numbered in sequence

Reviewer 4 Report

The paper is showing piloting effort on UF validation. I understood that such type of papers are in the scope of the journal

The paper describes the results at pilot at semi-industrial scale (20m3 /day) on the effect of air backwash on dead-end ultrafiltration of seawater to control the membrane fouling by using i) classical backwash (CB) and air backwash (AB) that consists to inject air in membrane module before a classical backwash. As experimental methodology the efficiency of AB and CB, the membrane resistances were used (e.g. membrane(Rm), reversible (Rrev) and irreversible (Rirr)). The study includes as cleaning monitoring tool the turbidity variation over time to identify the cleaning frequency. The novelty of the proposed backwash protocol as membrane cleaning is not innovative and the innovative part of the study is associated to the monitoring efforts on the cleaning performance. The paper is inside of the scope of the journal when considered as an application or case study.

The paper in its present format is having different flags and errors that should be carefully reviewed. The following issues should be addressed:

- It should be clarified the differences between air backwash and air scouring?

- In the introduction section it is not well described the UF cleaning options for operation, as additionally to the two options described nowadays forward flush, back flush and air scouring are used.

- Also it is needed a more deep review on the different fouling indexes typically used as cleaning monitoring with UF membranes?. Which are the limitations in comparison to the proposed methodology.

- The study describes as one variable of the study, the quality of the seawater, but not data on the composition of the seawater is provided. This point should be clarified?

- The composition of the sea water treated, especially the content on SS, TOC, SDI and TDI values should be provided. What are the different water qualities evaluated?

- The paper is not providing any statistical effort on data treatment, putting as example Figures 3 and 4 , where error bars are not provided.

- In equation 5, units of turbidity should be provided.

- The discussion of the paper should be improved and references to other studies should be published. An effort should be put on comparison with other cleaning monitoring tools? Or comparison of the evolution of the ratios of resistance values using other cleaning procedures.

- Why reversible values are not reported?. May be valuable information could be provided?

Other comments

Highlights: should be improved, are not describing properly the paper results

Keywords: avoid to use words already in the title.

Language: the text needs a review by a professional proof-reading system, additionally the text is prepared in a very technical way more suitable for an industrial report

Figures quality: needs attention as are figures with point sizes and size of the figure axes and figures captions are not easily readable. Specially Figure 7. However, Figure 7, is not having Figure caption.

Figure captions. should be improved in general. Figure captions should include all the information needed to be understood without reading the text.

Author Response

Referee 4: The paper is showing piloting effort on UF validation. I understood that such type of papers are in the scope of the journal

Thank you very much for your comment

The paper in its present format is having different flags and errors that should be carefully reviewed. The following issues should be addressed: It should be clarified the differences between air backwash and air scouring?

In agreement with the other referee comments, we have modified this part to better described our new air backwash

In the introduction section it is not well described the UF cleaning options for operation, as additionally to the two options described nowadays forward flush, back flush and air scouring are used.

We have no described this part because some references are given

Also it is needed a more deep review on the different fouling indexes typically used as cleaning monitoring with UF membranes?. Which are the limitations in comparison to the proposed methodology.

It is important to note that the UF is operated with constant permeate flux and in this case the fouling indexes is determined by the TMP measurement. We have added a sentence in agreement with your comment.

The study describes as one variable of the study, the quality of the seawater, but not data on the composition of the seawater is provided. This point should be clarified? The composition of the sea water treated, especially the content on SS, TOC, SDI and TDI values should be provided. What are the different water qualities evaluated?

We have studied during 9 months the UF filtration of natural sea water, it is impossible to determine on line these parameters for a semi industrial pilot plant versus time. Turbidity is the only parameter easy to record. We are agreed with you it is not perfect and we have written in the paper this problem. Moreover we have showed that whatever the quality of water the fouling resistance was the same.

The paper is not providing any statistical effort on data treatment, putting as example Figures 3 and 4 , where error bars are not provided.

We are agreed with your comment and we have added the error range in the text.

In equation 5, units of turbidity should be provided.

In agreement with your comment the unit is given

The discussion of the paper should be improved and references to otherstudies should be published. An effort should be put on comparison with other cleaning monitoring tools? Or comparison of the evolution of the ratios of resistance values using other cleaning procedures.

When you would like to compare some references, it is not very easy due to the quality of the feed. In this paper we would like to demonstrate that it is possible to estimate the fouling resistance with the variation of NTU versus time.

Why reversible values are not reported?. May be valuable information could be provided?

We don’t understand your comment, when the irreversible parameter is given the reversible values are also given.

Other comments

Highlights: should be improved, are not describing properly the paper results

In agreement with your comment the highlights are improved

Keywords: avoid to use words already in the title.

In agreement with your comment, we have modified/improved the keywords

Language: the text needs a review by a professional proof-reading  system, additionally the text is prepared in a very technical way more suitable for an industrial report

A native speaker improves the quality of the paper

Figures quality: needs attention as are figures with point sizes and size of the figure axes and figures captions are not easily readable. Specially Figure 7. However, Figure 7, is not having Figure caption. Figure captions. should be improved in general. Figure captions should include all the information needed to be understood without reading the text.

It is the same comment for all referees. In the case of word file, there is no problem. These errors appeared when the document was changed from word document to pdf

Round 2

Reviewer 1 Report

  1. Please check reference throughout the manuscript. “Error! Reference source not found” appears many times throughout the manuscript eg. Line 92, 134, 242 etc. Please check the PDF file that you have submitted.
  2. The Figure in the manuscript is still in a mess messed. Figure 2 appears in Pg 7-10. This make the manuscript very difficult to understand and follow.

 Please check the PDF version that you have submitted.

Author Response

Please check reference throughout the manuscript. “Error! Reference source not found” appears many times throughout the manuscript eg. Line 92, 134, 242 etc. Please check the PDF file that you have submitted.

We check reference throughout the manuscript

The Figure in the manuscript is still in a mess messed. Figure 2 appears in Pg 7-10. This make the manuscript very difficult to understand and follow.

 Please check the PDF version that you have submitted.

We submitted our paper in world file without PDF check (on membrane site). So for us is ok but not for you when you received a pdf file. It is not our fault. I have sent an email to the editor about this mistake. SORRY AGAIN. We will submit a good pdf file for the next submission.

Reviewer 2 Report

I’m EXTREMELY disappointed with the responses given by the authors. In fact, I see very LITTLE (or none) improvement in the manuscript. I would like to advise editor to go through the organization of the manuscript carefully. The authors DIDN’T revise based on the comments given. I remain my decision to reject this paper from publication!

I refer to my comments that were first given to authors.

(a) First of all, the use of dead-end UF membrane system for seawater treatment is NOT practical at all. Fouling is very severe and flux declined rapidly with the use of dead-end membrane. I don't know how the authors could produce convincing data (Figure 1) as a function of time since feed solution concentration is getting higher and higher as time proceeds. Besides, there is no indication for the data in Orange and Blue (Figure 1).

We are agreed with your comment. In fact we have specified in our paper that The seawater effluent used is located at Ifremer Station, Bouin (France) for which seawater received several pretreatments. We have added the detail of these pretreatments used (sand filtration and UV). We have modified the figure 1 to specify the orange and blue curves

New comment (21 March 2020) – The authors completely ignored my critical comments. Even with the pretreatment, fouling is still severe for the membrane. All of the large-scale desalination plants come with pretreatment processes, but fouling in the membrane is still one of the biggest challenges. The response given by the authors is completely UNACCEPTABLE! The authors used the dead-end UF membrane to run for 9000 min (~150 hours) and I wonder how the feed properties can be remained the same. They should show the feed properties as a function of time as well. If not, the results presented were badly affected by the variation of feed concentration!

All the figures are in fact not professionally prepared. Most of them are low resolution and unclear. Besides, there are 4 images in Figure 5, but there only 2 images (a and b) are described in the caption.

There is a mistake, only 2 images on figure 5???

New comment (21 March 2020) – In the PDF version (Page 6 of 12), four images are shown in Figure 5. If the authors did check the PDF file before clicking the submission, they sure can find the mistake.  In addition, no action was action to improve the figures technically. Figure 3 and 4 – For 60 LMH, data were recorded for 20, 30, 45 and 60 min while for 80 and 100 LMH, different times were recorded. Can the authors be consistent in presenting the data??? Figure 5 – The “NTU(min)” is placed beyond the image border. This should not happen for paper submission! Same mistake happened to other figures!

There are several figures between Figure 6 and 8, but they come without figure number and caption!!

We have modified all the figures in agreement with your remarks

New comment (21 March 2020) – There is no change AT ALL. In the PDF version (Page 7, 8, 9 and 10), there are NO caption for the image at ALL!!!!!

The selection of 200-kDa PES membrane for pretreatment is unclear. In most of the large-scale seawater desalination process, membrane with much smaller pore size is used!

Yes and no. Yes-We are agreed with your remark and now the pretreatment was defined and No- It is function of the pretreatment and it is for desalination (no our case). We have already studied the comparison between MF and UF for seawater filtration for other application and 20 nm of pore size is perfect.

Guilbaud J., Wyart Y., Moulin P. “ Economic viability of treating ballast water of ships by ultrafiltration as a function of the process position” Journal of Marine Science and Technology 12 (2018) 1-12 (DOI: 10.1007/s00773-018-0618-3)

Guilbaud J., Y. Wyart, K. Klaag and P. Moulin, Comparison of seawater and freshwater ultrafiltration on semi-industrial scale: ballast water treatment application, Journal of membrane science and research 4 (2018) 136-145

This pore size is now given in the membrane characteristics

New comment (21 March 2020) – The authors provided papers to justify why 200-kDa can be used. But, such references are less convincing. They should support the statement with papers publish in Journal of Membrane Science, Desalination and Separation and Purification Technology.

Throughout the manuscript, there is no any instrumental characterization on the membrane fouling. Such characterization is compulsory for high-quality paper publication.

It is impossible to characterize the membrane fouling because we used seawater with a high variation of the quality of this seawater. Moreover, the flow rate is very important. It is not a study at a lab scale.

New comment (21 March 2020) – Such response is completely UNACCEPTABLE! Since the authors said the seawater was pretreated, the samples will therefore exhibit similar properties. Thus, fouling characterization can actually be carried out. There are many papers reporting fouling characterization data using similar feed samples!

The properties of feed solution to PES membrane should be provided. Authors also need to provide what are the pretreatment processes conducted on the seawater before it was used for hollow fiber membrane treatment.

In agreement with your comment we have added these informations

New comment (21 March 2020) – There are no additional information about feed solution properties (pH, TSS, turbidity, conductivity, etc) after pre-treatment!

The separation data of PES membrane is completely missing! Such data is critical to correlate to the surface fouling!

We have added this information.

New comment (21 March 2020) – The authors are terrible! No such data shown!

Author Response

I’m EXTREMELY disappointed with the responses given by the authors. In fact, I see very LITTLE (or none) improvement in the manuscript. I would like to advise editor to go through the organization of the manuscript carefully. The authors DIDN’T revise based on the comments given. I remain my decision to reject this paper from publication!

Dear referee, I am sorry for our comments and our modifications and we will try to more explain, to more detail our comments and our results in the paper and in the response file.

I refer to my comments that were first given to authors.

(a) First of all, the use of dead-end UF membrane system for seawater treatment is NOT practical at all. Fouling is very severe and flux declined rapidly with the use of dead-end membrane. I don't know how the authors could produce convincing data (Figure 1) as a function of time since feed solution concentration is getting higher and higher as time proceeds. Besides, there is no indication for the data in Orange and Blue (Figure 1).

We agree with your comment. In fact, we have specified in our paper that the seawater effluent used is located at Ifremer Station, Bouin (France) and received several pretreatments. We have added the detail of these pretreatments used (sand filtration and UV). We have modified the figure 1 to specify the orange and blue curves

New comment (21 March 2020) – The authors completely ignored my critical comments. Even with the pretreatment, fouling is still severe for the membrane. All of the large-scale desalination plants come with pretreatment processes, but fouling in the membrane is still one of the biggest challenges. The response given by the authors is completely UNACCEPTABLE! The authors used the dead-end UF membrane to run for 9000 min (~150 hours) and I wonder how the feed properties can be remained the same. They should show the feed properties as a function of time as well. If not, the results presented were badly affected by the variation of feed concentration!

New answer (2 April 2020)

No, your comment is taken into account. For us everything is clear but perhaps not well explained. We agree with you:

Yes, dead end UF run during very long time and the fouling was severe and the feed properties changed. BUT for explain these points we have given an example: the figure 2.

Yes, the fouling is severe i.e. the strongly decrease of the permeability versus time is given even if one backwash is realized each 30 minutes, one air backwach each 180 minutes, one chemical wash even 1000 min! (around 2 per day). It is very important to note that the fouling is severe in our case as for large scale desalination plants, with a backwash each hour and 2 chemical washes per day. The variation of permeability versus time explains this severe fouling.

Yes, the feed quality changed and the variation of turbidity is given. Yes, the variation of the feed concentration affected the fouling: we have drawn 2 red lines to explain this specific point.

We have added in text some sentences to put in light these points.

All the figures are in fact not professionally prepared. Most of them are low resolution and unclear. Besides, there are 4 images in Figure 5, but there only 2 images (a and b) are described in the caption.

There is a mistake, only 2 images on figure 5???

New comment (21 March 2020) – In the PDF version (Page 6 of 12), four images are shown in Figure 5. If the authors did check the PDF file before clicking the submission, they sure can find the mistake.  In addition, no action was action to improve the figures technically. Figure 3 and 4 – For 60 LMH, data were recorded for 20, 30, 45 and 60 min while for 80 and 100 LMH, different times were recorded. Can the authors be consistent in presenting the data??? Figure 5 – The “NTU(min)” is placed beyond the image border. This should not happen for paper submission! Same mistake happened to other figures!

New answer (2 April 2020)

We submitted our paper in world file without PDF check (on membrane site). So, for us it’s ok but not for you when you received a pdf file. It is not our fault. I have sent an email to the editor about this mistake. SORRY AGAIN.

About the figures 3 and 4 and your new comment: In agreement with your first comment the fouling is severe so when the filtration time increases the duration decreases otherwise, we will have an irreversible fouling even by chemical wash. We have added an explanation in the paper.

There are several figures between Figure 6 and 8, but they come without figure number and caption!!

We have modified all the figures in agreement with your remarks

New comment (21 March 2020) – There is no change AT ALL. In the PDF version (Page 7, 8, 9 and 10), there are NO caption for the image at ALL!!!!!

New answer (2 April 2020)

The problem about the caption missing. For the figure 6, we understand now that you want a caption for each curve and we have added this caption.

The selection of 200-kDa PES membrane for pretreatment is unclear. In most of the large-scale seawater desalination process, membrane with much smaller pore size is used!

Yes and no. Yes-We are agreed with your remark and now the pretreatment was defined and No- It is function of the pretreatment and it is for desalination (no our case). We have already studied the comparison between MF and UF for seawater filtration for other application and 20 nm of pore size is perfect.

Guilbaud J., Wyart Y., Moulin P. “ Economic viability of treating ballast water of ships by ultrafiltration as a function of the process position” Journal of Marine Science and Technology 12 (2018) 1-12 (DOI: 10.1007/s00773-018-0618-3)

Guilbaud J., Y. Wyart, K. Klaag and P. Moulin, Comparison of seawater and freshwater ultrafiltration on semi-industrial scale: ballast water treatment application, Journal of membrane science and research 4 (2018) 136-145

This pore size is now given in the membrane characteristics

New comment (21 March 2020) – The authors provided papers to justify why 200-kDa can be used. But, such references are less convincing. They should support the statement with papers publish in Journal of Membrane Science, Desalination and Separation and Purification Technology.

New answer (2 April 2020)

We have added these references in agreement with the membrane used. For us the quality of the journal is important but the agreement with the membrane used is more important

Throughout the manuscript, there is no any instrumental characterization on the membrane fouling. Such characterization is compulsory for high-quality paper publication.

It is impossible to characterize the membrane fouling because we used seawater with a high variation of the quality of this seawater. Moreover, the flow rate is very important. It is not a study at a lab scale.

New comment (21 March 2020) – Such response is completely UNACCEPTABLE! Since the authors said the seawater was pretreated, the samples will therefore exhibit similar properties. Thus, fouling characterization can actually be carried out. There are many papers reporting fouling characterization data using similar feed samples!

New answer (2 April 2020)

We do not agree with your remark, as for membrane treatment, the quality of the water after the pretreatment (sand filtration (25-30 μm) and UV) is function of the feed water (seawater). During this study (one year) we have a large variation of all parameters of the sea water and also of all parameters of the pretreated water. It is impossible to record all these parameters.

The properties of feed solution to PES membrane should be provided. Authors also need to provide what are the pretreatment processes conducted on the seawater before it was used for hollow fiber membrane treatment.

In agreement with your comment we have added these pretreatment informations (sand filtration (25-30 μm) and UV)

New comment (21 March 2020) – There are no additional information about feed solution properties (pH, TSS, turbidity, conductivity, etc) after pre-treatment!

I am sorry but I can’t, during more than 9 months, recorded all the parameters of seawater, pretreated seawater and permeate: it is impossible, we can’t. We have added some average values for seawater.

The separation data of PES membrane is completely missing! Such data is critical to correlate to the surface fouling!

We have added this information.

New comment (21 March 2020) – The authors are terrible! No such data shown!

New answer (2 April 2020):

Sorry but in agreement with your remark and the parameter studied (turbidity) a sentence was added : About the membrane performance, a turbidity lower than 1 NTU is obtained in the permeate.

Reviewer 3 Report

Does the "Cleaning efficiency higher than 100%" mean the membrane was destroyed?

Author Response

Does the "Cleaning efficiency higher than 100%" mean the membrane was destroyed?

No, in agreement with the equation 5, higher than 100% explains a better backwash efficiency than the previous one.

Reviewer 4 Report

The new version of the manuscript is not claiming some of the requested changes:

As an example

Figures numbering. There is Fgure 8 and there is not Figure 7

Fugures ordering. After Figure 1 is Figure 3, and then Figure 2 is between figure 6 and 7. 

Authors should provide answers to all the requests: as an example

At the end of my report was written

The Figure ordering/numbering in the manuscript is a mess. In Pg 10  appears Figure 2. This make the manuscript very difficult to understand and follow.

Author Response

The new version of the manuscript is not claiming some of the requested changes:

As an example

Figures numbering. There is Fgure 8 and there is not Figure 7

Fugures ordering. After Figure 1 is Figure 3, and then Figure 2 is between figure 6 and 7.

Authors should provide answers to all the requests: as an example

At the end of my report was written

The Figure ordering/numbering in the manuscript is a mess. In Pg 10  appears Figure 2. This make the manuscript very difficult to understand and follow.

It is not our fault. We submitted our paper in Word file without PDF check (on membrane site). So, for us it is ok but not for you when you received a pdf file. I have sent an email to the editor about this mistake. SORRY AGAIN. We will submit a good pdf file for the next submission.

Round 3

Reviewer 1 Report

  1. I would suggest to remove cross-referencing in the text. “Error! Reference source not found” appears many times throughout the manuscript eg. Line 92, 144, 195, 254 in PDF file. Please check the PDF file that you have submitted.
  2. Line 186 “data point” should be used instead of “experience”?
  3. Line 187, “experiment” is more appropriate instead of “experience”
  4. The legend of Figure 5 should be made consistent. In Figure 5a, the legend indicates AB/AC frequency 1/3, but in Figure 5b, the legend is J and filtration time. Try to make both consistent
  5. Typo in caption of Figure 5, “…and (b): [AB/AC frequency: 1/5; J =60L.h-1.m-2 et tfiltration 30min]”. The “et” in the caption should be “and”.
  6. The statement “ The reversible resistance remains constant” made in line 234 is not true. The reversible resistance is not constant but rather increasing at a slower pace as compared to R irr.

Author Response

Thank you very much for your comments. We have appreciated the revisions and have made the corrections and changes that they have suggested. You will find hereafter our responds identifying the modifications made. We hope that all these corrections, in light of the remarks of the referee, can make the manuscript publishable in Membranes

I would suggest to remove cross-referencing in the text. “Error! Reference source not found” appears many times throughout the manuscript eg. Line 92, 144, 195, 254 in PDF file. Please check the PDF file that you have submitted.

We have removed all “Error! Reference source not found” and check the pdf file. Sorry again

Line 186 “data point” should be used instead of “experience”?

Line 187, “experiment” is more appropriate instead of “experience”

We have modified these points in agreement with your comment

The legend of Figure 5 should be made consistent. In Figure 5a, the legend indicates AB/AC frequency 1/3, but in Figure 5b, the legend is J and filtration time. Try to make both consistent

Typo in caption of Figure 5, “…and (b): [AB/AC frequency: 1/5; J =60L.h-1.m-2 et tfiltration 30min]”. The “et” in the caption should be “and”.

We have modified these points

The statement “ The reversible resistance remains constant” made in line 234 is not true. The reversible resistance is not constant but rather increasing at a slower pace as compared to R irr.

We have modified this sentence in agreement with your comment

Reviewer 2 Report

To editor/author,

Page 3 -  Equation number is missing for the integral turbidity.   Figure 1 - Label should be given to the photo as well.   Figure 3 and 4 - The authors are strongly advised to use SAME pattern (including colour and bar pattern) to present the data.

Figure 5 -  Same font size and image dimension should be used for both (a) and (b). Besides, legends should be placed within X- and Y-axis.

Figure 7 - Same font size and image dimension should be used for all images. Besides, the authors still used "E" for the y-axis!. I have raised this issue before! The caption of the figure should be revised to reflect the images correctly.

Author Response

Thank you very much for your comments. We have appreciated the revisions and have made the corrections and changes that they have suggested. You will find hereafter our responds identifying the modifications made. We hope that all these corrections, in light of the remarks of the referee, can make the manuscript publishable in Membranes

Page 3 -  Equation number is missing for the integral turbidity.

In agreement with an other reviewer, we have removed this number because it is not an equation but a relation

Figure 1 - Label should be given to the photo as well.

We have specified Figure 1a et Figure 1b

Figure 3 and 4 - The authors are strongly advised to use SAME pattern (including colour and bar pattern) to present the data.

We have used the same pattern in agreement with your comment

Figure 5 -  Same font size and image dimension should be used for both (a) and (b). Besides, legends should be placed within X- and Y-axis.

Figure 7 - Same font size and image dimension should be used for all images. Besides, the authors still used "E" for the y-axis!. I have raised this issue before! The caption of the figure should be revised to reflect the images correctly.

We have modified these figures, we did the best we could.